# Blood–Brain Barrier Biomarkers before and after Kidney Transplantation

**DOI:** 10.3390/ijms24076628

**Published:** 2023-04-01

**Authors:** Leah Hernandez, Liam J. Ward, Samsul Arefin, Peter Barany, Lars Wennberg, Magnus Söderberg, Stefania Bruno, Vincenzo Cantaluppi, Peter Stenvinkel, Karolina Kublickiene

**Affiliations:** 1Division of Renal Medicine, Department of Clinical Science, Intervention and Technology (CLINTEC), Karolinska Institutet, 171 77 Stockholm, Sweden; 2Department of Forensic Genetics and Forensic Toxicology, National Board of Forensic Medicine, 587 58 Linköping, Sweden; 3Department of Transplantation Surgery, Karolinska University Hospital, 141 86 Stockholm, Sweden; 4Department of Pathology, Clinical Pharmacology and Safety Sciences, R&D AstraZeneca, 431 83 Gothenburg, Sweden; 5Department of Medical Sciences, University of Torino, 10124 Torino, Italy; 6Nephrology and Kidney Transplant Unit, Department of Translational Medicine (DIMET), University of Piemonte Orientale (UPO), “Maggiore della Carita” University Hospital, 28100 Novara, Italy

**Keywords:** CKD, end-stage kidney failure, kidney transplantation, BBB, NSE, NfL, BDNF, extracellular vesicles

## Abstract

Kidney transplantation (KT) may improve the neurological status of chronic kidney disease (CKD) patients, reflected by the altered levels of circulating BBB-specific biomarkers. This study compares the levels of neuron specific enolase (NSE), brain-derived neurotrophic factor (BDNF), neurofilament light chain (NfL), and circulating plasma extracellular vesicles (EVs) in kidney-failure patients before KT and at a two-year follow up. Using ELISA, NSE, BDNF, and NfL levels were measured in the plasma of 74 living-donor KT patients. Plasma EVs were isolated with ultracentrifugation, and characterized for concentration/size and surface protein expression using flow cytometry from a subset of 25 patients. Lower NSE levels, and higher BDNF and NfL were observed at the two-year follow-up compared to the baseline (*p* < 0.05). Male patients had significantly higher BDNF levels compared to those of females. BBB biomarkers correlated with the baseline lipid profile and with glucose, vitamin D, and inflammation markers after KT. BBB surrogate marker changes in the microcirculation of early vascular aging phenotype patients with calcification and/or fibrosis were observed only in NSE and BDNF. CD31+ microparticles from endothelial cells expressing inflammatory markers such as CD40 and integrins were significantly reduced after KT. KT may, thus, improve the neurological status of CKD patients, as reflected by changes in BBB-specific biomarkers.

## 1. Introduction

Chronic kidney disease (CKD) is a global health burden [1,2,3] and a model of premature or accelerated aging [4,5]. The final stage of CKD, end-stage kidney failure (ESKF), requires renal replacement therapy (RRT) in the form of dialysis or kidney transplantation (KT) to enhance patient survival, with KT as the preferred RRT for ESKF [6]. Patients with ESKF are more susceptible to cardiovascular disease (CVD) [7] and central nervous system (CNS) complications such as strokes, cognitive impairment, depression, anxiety, seizures, and encephalopathy [8,9,10], which increases morbidity and mortality in this population. CKD-induced CNS disorders may be due to the retention of toxic uremic metabolites that mediate chronic inflammation, oxidative stress, endothelial dysfunction, vascular calcifications, and compromise the blood–brain barrier (BBB) [10,11] that may manifest as altered mental status or a physical disability [8]. Increased mortality among ESKF patients is brought about by the elevated levels of uremic toxins and persistent inflammatory response causing systemic vascular structural, functional, and histological damage [12]. Furthermore, persistent oxidative stress, inflammation, and metabolic disturbances in CKD may trigger the release of extracellular vesicles (EVs). These microparticles are involved in cell-to-cell communication through the transfer of bioactive proteins, lipids, and genetic material such as mRNA, miRNA, and lncRNA [13,14]. EVs may further impact the progression of CKD and the early vascular aging phenotype (EVA). These microparticles carry the molecular profile of parent cells in the uremic environment [15], induce endothelial dysfunction [16], and can pass through the BBB due to enhanced permeability, thus reaching neurons and glial cells inducing direct damage [17,18].

The BBB is a specialized network of arteries surrounding the brain that protect it from toxic circulatory molecules [19]. The uremic environment may disrupt BBB permeability through the disruption of tight junction proteins [20]. Consequently, harmful substances may pass through the barrier, causing toxins to accumulate in the brain while simultaneously releasing BBB-specific compounds that can serve as indicators of the pathological process in the BBB. Mild vascular cognitive impairment is related to increased BBB permeability, indicating that BBB breakdown contributes to cognitive decline, as demonstrated by increased leakage in magnetic resonance imaging and a lower Montreal Cognitive Assessment score [21]. The levels of brain-specific circulating biomarkers such as NSE, which is a biomarker for neuronal cell death [22], and BDNF, which regulates the survival and maintenance of neurons [23], are increased in hemodialytic patients compared to the controls [20]. Alterations in NSE levels suggest changes in the brain, and high levels of NSE can trigger neuroinflammation, extracellular matrix degradation, glial-cell proliferation, and actin remodeling that could affect the macrophage and microglia migration, thus promoting neuronal cell death [24].

In ESKF patients, it is crucial to determine the neurological status since it is linked to daily living activities and other social aspects that may affect the patient’s quality of life. Circulating brain-specific biomarkers are useful indicators of the possible amelioration of neurological status through the improvement of BBB permeability in ESKF. Besides NSE and BDNF, NfL is another protein expressed in neurons [25,26] that is released into the CSF and blood after damage to the CNS and peripheral nervous system (PNS) [25,26]. NfL is a potential marker of senescence [25,26], although the exact mechanisms by which NfL is released from neuroaxonal injury are not fully understood. Elevated levels may be a direct result of cell integrity loss [27].

To explore the effect of KT on BBB, we compared the levels of three blood–brain-specific biomarkers and determined if there were sex-specific changes in BBB biomarkers after KT. Moreover, we evaluated the effect of KT on EV size, concentration, and surface antigen expression, primarily focusing on endothelial-cell-derived microparticles. As KT may improve the neurologic state of ESKF patients, we hypothesized that this would be reflected in the altered levels of circulating BBB-specific biomarkers and EVs. BBB integrity was assessed with the peripheral biomarkers whose levels may differ significantly before and after KT. 

## 2. Results

### 2.1. Characteristics of the Study Population

Table 1 summarizes the clinical features of the KT patients. The baseline median age was 46 years, and the population was composed of 70% male participants. Most patients had CVD (84%), and only 8% had DM. Results show that, two years after KT, there were improvements in BMI, SBP, creatinine, albumin, phosphate, troponin T, HDL, Lp(a), and homocysteine. Additionally, there was a significant reduction in the use of Ca channel blockers, but there was increased utilization of ACEi/ARB and statins.

#### Characteristics of Male and Female Participants at Baseline

Males had higher BMI, SBP, DBP, and creatinine compared to those of females. In addition, more males than females were on calcium channel blocker medications. Females also had higher HDL and apo-A1 than those of males (Appendix A). 

### 2.2. Circulating Biomarker Levels

Results of the circulating biomarkers show a reduction in NSE levels (3.4 ng/mL, IQR 2.6–4.6 vs. 4.6 ng/mL, IQR 3.5–5.8) two years after KT, while BDNF (1.3 ng/mL, IQR 0.8–2.0 vs. 1.1 ng/mL, IQR 0.8–1.5) and NfL (129.7 ng/mL, IQR 104–162.9 vs. 92 ng/mL, IQR 70.2–108.4) levels were increased at two years post-KT compared to the baseline. Interassay CV values were 4% for NSE and NfL, and 9% for BDNF (Figure 1).

### 2.3. Sex-Disaggregated Analysis of Biomarkers

The sex-divided analysis of BBB biomarkers indicates a significantly reduced trend in the NSE levels of both males (3.4 ng/mL, IQR 2.6–4.8 vs. 4.5 ng/mL, IQR 3.4–6.0) and females (3.5 ng/mL, IQR 2.5–3.9 vs. 4.7 ng/mL, IQR 3.8–5.7) after KT. For BDNF, there were significantly increased levels of BDNF in males (1.5 ng/mL, IQR 0.8–2.3 vs. 1.1 ng/mL, IQR 0.7–1.4), but not in females (1.1 ng/mL, IQR 0.7–1.5 vs. 1.2 ng/mL, IQR 0.7–1.5). NfL results also show significantly increased levels in both males (138.2 ng/mL, IQR 110.4–173.1 vs. 94.7 ng/mL, IQR 67.0–110.5) and females (113.2 ng/mL, IQR 99.7–144.8 vs. 86.5 ng/mL, 70.2–108.22) after KT (Figure 1). 

The analysis of the difference in the peripheral biomarkers between males and females only showed significant differences in BDNF levels at two-year follow-up, with males having higher BDNF levels compared to those of females (Table 2).

### 2.4. Correlation Analysis

At baseline, biomarkers were primarily associated with lipid profile. BDNF was negatively associated with troponin T (rs = −0.33, *p* = 0.005, n = 74), NfL was negatively correlated with HDL (rs = −0.23, *p* = 0.044, n = 74), and NSE was positively correlated with cholesterol (rs = 0.40, *p* < 0.001, n = 74), apo-B, and ApoB1 (rs = 0.26, *p* = 0.023, n = 74) (Figure 2).

After KT, positive and negative associations were observed not only with the lipid profile, but also with glucose, vitamin D, and inflammation, indicating the importance of controlling the metabolic environment post-KT in relation to the BBB markers.

### 2.5. BBB Biomarker Levels According to Calcification and Fibrosis Score in Epigastric Artery

The levels of the biomarkers according to calcification show that NSE levels were significantly reduced in both patients with (3.7 ng/mL, IQR 2.8–4.8 vs. 4.7 ng/mL, IQR 3.5–6.0) and without (2.9 ng/mL, IQR 2.3–3.9 vs. 4.9 ng/mL, IQR 3.8–6.8) calcification. The results for increased BDNF (1.5 ng/mL, IQR 0.9 ng/mL, IQR 0.6–1.6) post-KT, on the other hand, were significant only in patients with no calcification. NfL demonstrated a significant increase in levels after KT both in patients with (132.8 ng/mL, IQR 101.9–186.3 vs. 93 ng/mL, IQR 70.9–107.3) and without (122.1 ng/mL, IQR 109.1–160.1 vs. 81.2 ng/mL, IQR 66.1–96.1) calcification (Figure 3).

In biomarker levels according to fibrosis score, NSE (3.2 ng/mL, IQR 2.5–4.4 vs. 4.9 ng/mL, IQR 3.8–6.3) showed similar reduced levels after KT compared to those at the baseline among patients without fibrosis. Only those patients without fibrosis had a significant rise in BDNF (1.5 ng/mL, IQR 1.0–2.1 vs. 1.1 ng/mL, IQR 0.6–1.4). NfL levels show a significant increase two years after KT in the groups both with (132.8 ng/mL, IQR 105.7–176.6 vs. 94.4 ng/mL, IQR 75.5–104.5) and without (125.1 ng/mL, IQR 107.5–160.9 vs. 84.4 ng/mL 66.4–102.7) fibrosis (Figure 4). 

When both calcification (Score 1–3) and fibrosis (Score 1–3) were present in patients, results showed significant changes only in the NfL biomarker (138 ng/mL, IQR 116.1–196.9 vs. 97.2 ng/mL, IQR 84.9–108.6), and not in NSE and BDNF (Figure 5).

### 2.6. Plasma EV Phenotypic Characterization

Plasma EV isolation and phenotypic characterization were performed in 25 representative patients of the whole ESKF cohort at the baseline and 2 years after KT. Nanosight analysis showed a significant decrease in plasma EV concentration after KT without identifying any difference in size (160–180 nm on average at both considered time points; Figure 6A).

The flow cytometry analysis of surface protein expression revealed a significant reduction in endothelial antigen CD31, costimulatory molecule CD40, alpha 2 and 5, and beta 1 integrins (Figure 6B). Results suggest that KT may modulate the release of EVs via inflamed and dysfunctional endothelial cells, a typical biological condition of ESKF patients.

## 3. Discussion

The present study shows that KT may benefit BBB integrity, as evidenced by the positive changes in circulating BBB-specific biomarkers NSE and BDNF. We also observed favorable changes in relevant parameters to increased cardiovascular risk. We report a link between BBB biomarkers and the metabolic profile of ESKF patients. Sex disaggregated analysis shows that males had significantly higher levels of BDNF compared to those of females. The severity of the EVA phenotype in microcirculation was reflected in the changes in NSE and BDNF, but not in NfL. Lastly, to further confirm the favorable changes in variables associated with an increased cardiovascular risk, we found a significant decrease in CD31+ endothelial-cell-derived EVs after KT had expressed markers of inflammation and cell damage such as CD40 and different integrins. 

Because KT recipients may only have a single functioning kidney, they may still be regarded as having CKD and thus remain in an inflammatory state. Nevertheless, our study suggests that BBB status in ESKF patients may be improved following KT, as demonstrated by positive changes in the clinical indicators and levels of BBB biomarkers NSE and BDNF. Additionally, a previous report [28] suggested that KT could improve cognitive impairment in ESKF patients.

Two years post-KT, NSE levels were reduced, while BDNF and NfL levels increased. Thus, our study confirmed that improved renal function lowers NSE levels [29] and increases BDNF levels [30]. Contrary to our expectations, NfL levels increased after KT, possibly indicating NfL as a potential marker for senescence and subclinical CNS impairment [26]. Furthermore, as the patients in this study exhibited a phenotype of early vascular aging [4,31,32] with increased CVD comorbidity, elevated levels of NfL [33] after KT may reflect this.

The sex differences were apparent only in BDNF two years post-KT: male participants had higher levels of BDNF than those of females after two years, which is consistent with a previous meta-analysis [34]. As BDNF levels are linked to the hormonal state, postmenopausal women have significantly reduced levels [35]. The circadian rhythm may also affect BDNF levels, with the highest levels occurring during periods of activity or stress, and the lowest levels during rest periods [35,36]. Factors such as fasting [37] and medications, including psychoactive, lipid-lowering, antidiabetic, antihypertensive, and anti-inflammatory drugs, all widely prescribed in ESKF, may also affect BDNF levels [38]; therefore, further studies are warranted

Reduced renal function is a major risk factor for CVD [39], and controlling for associated risk factors is important. CKD and components of metabolic syndrome are interlinked [40,41,42,43], and the activation of signaling pathways involved in oxidative stress, inflammation, and endothelial dysfunction may play a role [40,41]. Low-grade chronic inflammation, dyslipidemia, vitamin D insufficiency, anemia, and uremic toxin accumulation contribute to metabolic syndrome in CKD [41]. KT improves and/or corrects a variety of cardiovascular and metabolic risk factors [44], but post-KT variables such as immunosuppressive regimens, infection, the retention of uremic toxins, and KT itself [44] may still increase the risk. These could partially explain the observed correlation between BBB markers and certain laboratory parameters reflecting the metabolic profile of ESKF patients. 

Metabolic-syndrome-associated mediators can induce inflammation and microvascular changes that could result in structural damage to the CNS and impair cognition [45,46]. The components of metabolic syndrome affect the brain of adolescents and adults alike [47]. The affected domains include memory deficit, visuospatial abilities, executive function, speed processing, and overall intellectual function [47]. Thus, it is not surprising that components of metabolic syndrome might influence surrogate markers of BBB. BBB biomarkers are correlated with components that increase the risk for CVD and metabolic syndrome. Elevated NSE levels may reflect both neurologic impairment [24] and metabolic syndrome [48], while reduced levels of BDNF were observed in patients with metabolic syndrome [34,45]. NfL, on the other hand, reflects small-vessel disease pathology on MRI and cognitive tests [49], further reinforcing its connection to metabolic syndrome and increased CVD risk in our ESKF patients. 

When examining signs of the EVA phenotype, surrogate markers of BBB did not reflect macrocirculation changes (i.e., epigastric artery). After KT, patients without calcification and fibrosis had favorable alterations in NSE and BDNF levels, but not in NfL. Patients with calcification and/or fibrosis, representing an early vascular ageing phenotype, did not have the predicted improvement in BBB status that would be reflected by favorable changes in the circulating levels of NSE and BDNF. 

Another interesting finding of this study is the modulation of EV plasma concentration and phenotype observed after KT. The involvement of microparticles in the mechanisms of ESKF-associated endothelial dysfunction, arterial stiffness, and increased cardiovascular risk of this frail population has recently been demonstrated. In particular, endothelial-cell-derived EVs are correlated with a loss of flow-mediated dilation, augmented aortic pulse-wave velocity, and an increased common carotid augmentation index [50]. Moreover, plasma EVs isolated from ESKF patients could promote the calcification of vascular smooth muscle cells (VSMC). The procalcific capacity of plasma EVs was significantly reduced after KT. This biological effect was mainly ascribed to the decreased expression of fetuin-A and MGP, and to the upregulation of annexin-A2 in EVs [51]. Uremic plasma EVs, mainly derived from endothelial cells and platelets, carry molecules involved in inflammation, immune response, apoptosis, and the triggering of coagulation and complement cascades such as CD40Ligand, ICOS, Fas ligand, tissue factor, and C5b9. These EVs are internalized in vascular cells, inducing the endothelial dysfunction and osteoblast differentiation of VSMC [16]. We observed a significant reduction in CD31+ endothelial EVs two years after KT. Furthermore, the downregulation of costimulatory molecule CD40, integrins alpha 2 and 5, and beta 1 was noticed. CD40, a 50 kDa integral membrane protein of the tumor necrosis factor receptor family, and its cognate agonist, CD40Ligand (CD40L or CD154), are coexpressed by several nonimmune cells, including endothelial cells, smooth muscle cells, and macrophages. A prospective observational multicenter trial investigated the relationship among chronic inflammation, morbidity, and mortality in 757 HD patients, and showed that higher sCD40L serum levels than 7.6 ng/mL were associated with increased cardiovascular risk [52]. Moreover, both CD40 and CD40L are highly expressed in calcified plaques, suggesting a relevant contribution of this pathway in the inflammatory status of atherosclerotic lesions [53]. An experimental therapeutic approach aimed to block the CD40–CD40L pathway using a monoclonal antibody that protects from the development and progression of atherosclerotic lesions with the reduced thickness of the aortic wall [54].

The integrin protein family plays a pivotal role in different biological activities, including blood clot formation, cell attachment, and migration, affecting two main aspects of acute inflammation: vascular permeability and leukocyte recruitment [55]. Increasing evidence suggests that pericytes play a crucial role in vascular remodeling: during inflammation, cytokines trigger a switch in pericyte integrins from alpha 1 to 2, stimulating their proliferation and migration on collagen, a fundamental mechanism of remodeling [56]. Integrin alpha 5 plays a key role in the pathological vascular remodeling associated with atherosclerosis through binding to fibronectin, which can sensitize endothelial cells to inflammatory stimuli [57]. Moreover, alpha 5 integrin binds to phosphodiesterase-4D5 (PDE4D5), which induces the PP2A-dependent dephosphorylation of PDE4D5 on inhibitory site Ser651: the in vivo knockdown of PDE4D5 inhibited inflammation in atherosclerotic plaques [58]. Beta-1 integrin can be released by hepatocytes under lipoxic stress as the cargo of EVs mediating monocyte adhesion to liver sinusoid endothelial cells, thus inducing the hepatic inflammation and progression of nonalcoholic steatohepatitis (NASH) [59]. In addition, galectin-3 can exacerbate oxidized LDL-induced endothelial injury by inducing inflammation through integrin beta1–RhoA–JNK signaling activation [60]. 

The results of this study were obtained from the same patients at different time points. Individual physiology can change over time, which can result in variations in sample measurements. Physiological changes may occur at the structural, functional, and molecular levels [61]. Additionally, preanalytical conditions at the two time points, such as posture, time of collection, food intake, exercise, and medication use, may affect the obtained results [62]. For example, the overall health of transplant patients can vary depending on factors such as the presence of infections, shifts in medication dosage and drug type, and immune status [63]. Creatinine, which is typically measured in CKD patients, can be affected by hydration status [64], medications, muscle mass, and nutrition [65,66]. In the field of personalized medicine, establishing an individual reference interval is also a challenge [67], especially for novel biomarkers. The interpretation of clinical laboratory test results typically involves a comparison with a reference interval that can serve as a clinical decision point for physicians [67]. Reference values may be subject to variation and influenced by the study population or the employed laboratory methods [67]. Therefore, a challenge may arise when there are no available existing reference values for a particular study population, hindering comparing the results with reference intervals that were set for a different population group. As far as we know, there are currently no universally accepted reference values for our chosen BBB markers. Due to the study design, our analysis was limited to the comparison of the biomarker levels at the baseline and two years post-KT using a predetermined *p*-value to assess significant changes. Temporal profiles, variations in handling, storage, or experimental conditions may also influence the physical or physiological properties of samples used in time-dependent investigations. The concentrations and measurements of certain analytes are affected by suboptimal temperatures [68,69] and the presence of unrecognized hemolysis [70]. The temporal profile of BBB biomarkers in CKD patients requires further investigation.

Our study’s strength is the longitudinal assessment by analyzing samples from the same patients at baseline and two years post-KT. However, since most of our participants were males, it is recommended that future research be conducted with a balanced sample of both sexes. We plan to use an in vitro model to demonstrate the functional characteristics of BBB permeability in a uremic environment. We also plan to investigate the relationship between Evs in a uremic environment, the worsening of the EVA phenotype, and BBB dysfunction in CKD. Indeed, the focus on endothelial-cell-derived microparticles may limit the evaluation of the complex biological activities of circulating plasma Evs, those derived from platelets and leukocytes in a clinical context of immunosuppression such as KT. 

## 4. Materials and Methods

### 4.1. Study Population

Biomarker analyses from the circulation were conducted on plasma samples from 74 living-donor KT patients [71,72] recruited between September 2012 and March 2019 for the baseline, and followed up two years post-KT between October 2014 and May 2021. 

Clinical data were extracted from the medical records of Karolinska Hospital patients. Demographic data were age, sex, BMI, the presence of comorbid conditions such as cardiovascular disease (CVD) and diabetes mellitus (DM), the use of antihypertensive medications ACE inhibitors (ACEi)/angiotensin receptor blockers (ARBs), beta blockers, and statins. Physicians diagnose CVD on the basis of the clinical symptoms of ischemic heart disease, peripheral vascular disease, and/or cerebrovascular disease [5]. 

### 4.2. Blood Plasma Biomarkers and Laboratory Analyses

Plasma samples from KT patients were obtained at the baseline and two years after KT, and kept at −80 °C before peripheral biomarker analysis. Commercially available enzyme-linked immunosorbent assay (ELISA) kits were used to quantify the circulating levels of the biomarkers of NSE (DENL20, R&D systems, Abingdon, UK), BDNF (DBD00, R&D systems, Abingdon, UK), and NfL (EKX-RPTZML-96, Nordic Biosite AB, Täby, Sweden) in the plasma samples, and used according to the manufacturer’s instructions. To accommodate the sample size, plasma samples were run as singlets for all kits. Other laboratory parameters, such as the lipid profile (triglyceride, cholesterol, HDL, apo-A1, apo-B, Lp(a)), creatinine, albumin, hsCRP, calcium, phosphate, troponin T, glucose, HBA1c, and IL-6 were examined using routine methods at the Department of Laboratory Medicine at Karolinska University Hospital [73,74,75,76].

### 4.3. EV Isolation from Plasma Samples

EVs were purified with plasma and characterized as previously described [16] according to the criteria suggested by the International Society for Extracellular Vesicles (ISEV) 2018 guidelines [77]. Briefly, blood was first centrifuged for 20 min at 6000× *g* to remove cells, apoptotic bodies, other large particles, and aggregates. Plasma aliquots were maintained at −80 °C until use and then further centrifuged to isolate EVs, which were enriched wuth ultracentrifugation for 2 h at 100,000× *g* at 4 °C using the SW60Ti rotor in a Beckman Coulter Optima L-90 K ultracentrifuge (Beckman Coulter, Fullerton, CA, USA). After supernatant removal, pellet was resuspended in DMEM for phenotype analysis.

### 4.4. Nanosight Analysis

Plasma EV preparations were diluted (1:1000) in sterile 0.9% saline, and analyzed with NanoSight LM10 (Nanosight, Amesbury, UK) equipped with Nanoparticle Analysis Systems and NTA 1.4 analytical software to evaluate EV size and concentration. Acquisitions were performed at a camera level setting of 14, and three 30 s videos were recorded for each sample: the number of total EVs (NTA/mL) was obtained by multiplying the value given by the instrument for the dilution for analysis by the ml amount in which EVs were resuspended [78,79]. 

### 4.5. Phenotypic Characterization of EVs

Plasma-derived EVs were subjected to bead-based multiplex analysis via flow cytometry (MACSPlex Exosome Kit, human, Miltenyi Biotech, Auburn, CA, USA) in accordance to ISEV guidelines. The MACSPlex kit allowed for detecting the expression of 39 surface antigens with different antibody-coated bead subsets. Briefly, the different EV preparations were diluted in a MACSPlex buffer to a final volume of 120 µL; then, 15 µL of MACSPlex exosome capture beads was directly added to the suspension. To detect the bead-linked Evs, APC-conjugated anti-CD9, anti-CD63, and anti-CD81 detection antibodies were added for 1 h in conditions of gentle oscillatory rotation at 450 rpm protected by light exposure. After incubation, unconjugated beads were washed 3 times at 3000× *g* for 5 min with the MACSPlex buffer; the supernatant was then carefully aspirated, leaving 150 µL for each sample for acquisition. Flow cytometric analysis was performed with a CytoFLEX flow cytometer (Beckman Coulter, Indianapolis, IN, USA) recording about 5000–8000 single bead events. The median fluorescence intensity (MFI) of all 39 exosome markers was corrected for the background and gated on the basis of their respective fluorescence intensity in accordance to manufacturer’s instructions [14,16]. Among the 39 antigens analyzed using the MACSPlex Exosome Kit, CD40, CD31, alpha 2 and 5, and beta 1 integrins are presented on the basis of significant (*p* < 0.05) observed findings.

### 4.6. Determination of Calcification and Fibrosis Scores

To evaluate and calculate the calcification and fibrosis scores, biopsies were taken from the epigastric arteries of KT patients during surgery. An experienced pathologist then evaluated the level of calcification and fibrosis, categorizing it as none (0), mild (1), moderate (2), or extensive (3) [71]. 

### 4.7. Statistical Analysis

The clinical and biochemical characteristics, and biomarker measurements are expressed as the median and interquartile range (IQR). Statistical analysis was conducted using GraphPad Prism (v9, GraphPad, San Diego, CA, USA) and SPSS (v28, IBM, New York, NY, USA). Shapiro–Wilk normality tests were used to assess the data distribution, and all subsequent analyses were conducted in line with the data distribution. To assess the changes in BBB biomarkers, EVs, and laboratory parameters within the same patient, the Wilcoxon paired test was used to analyze the differences between the baseline and 2-year follow-up data. Fisher’s test was used to analyze the categorical variables, and the Mann–Whitney U test was used to compare differences in the continuous variables between males and females for nonparametric analysis. To determine the association between BBB biomarkers and patient-related factors, such as demographic and clinical characteristics, correlation analysis was performed using Spearman’s rank correlation. Statistical significance was determined at *p* < 0.05.

## 5. Conclusions

This study yielded the promising finding that KT has favorable influence on the BBB microcirculation. The results demonstrated a link between the BBB biomarkers and metabolic profiles of ESKF patients. The severity of the EVA phenotype was reflected in the changes in specific surrogate biomarkers, underscoring the need for individualized treatment strategies. Overall, these findings suggest that KT leads to improved BBB status, as shown by the favorable alterations in BBB biomarker levels, particularly in patients without calcification and fibrosis. 

## Figures and Tables

**Figure 1 ijms-24-06628-f001:**
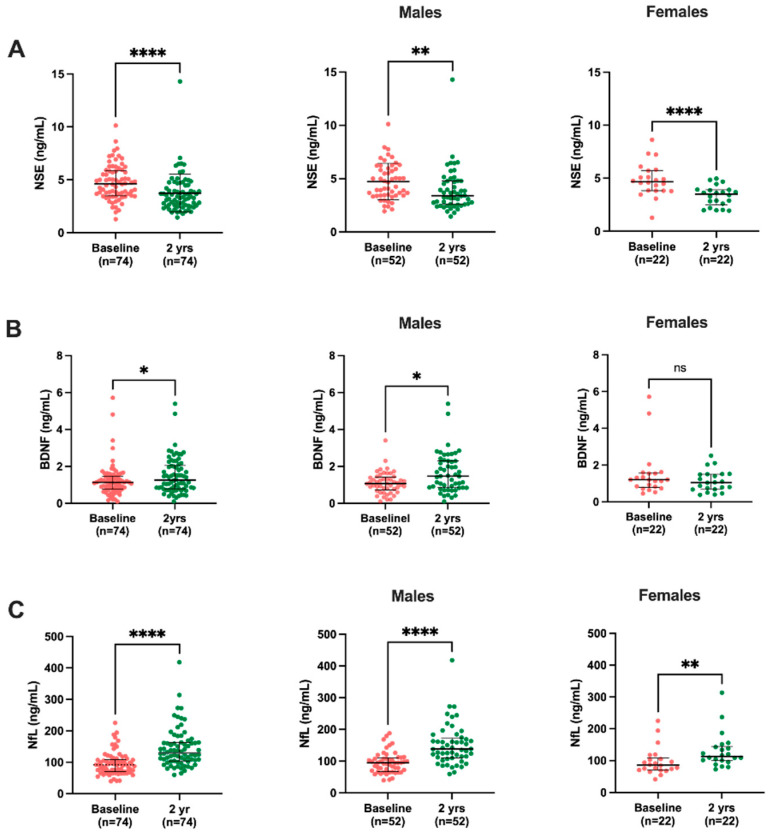
Comparison of biomarker (**A**) NSE, (**B**) BDNF, and (**C**) NfL levels at the baseline and two-year follow-up among KT patients. Results are presented as the median and IQR. The differences between baseline and two years post-KT were analyzed using the Wilcoxon paired test. * *p* < 0.05, ** *p* < 0.01, **** *p* < 0.0001.

**Figure 2 ijms-24-06628-f002:**
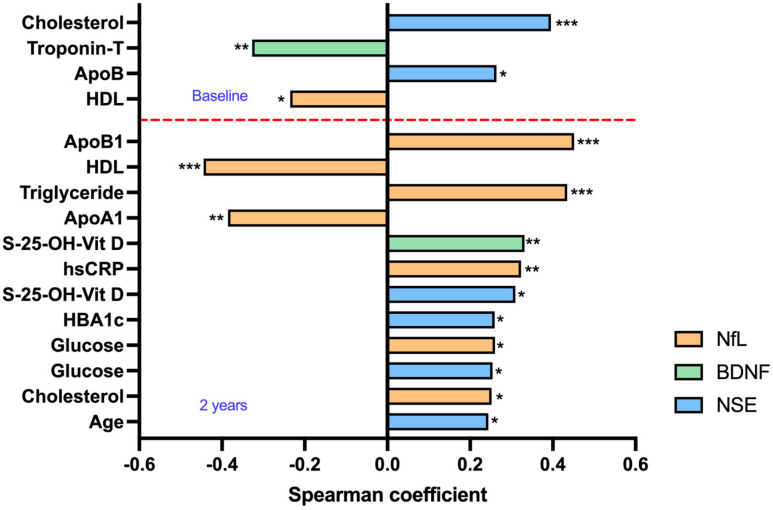
Correlation of biomarkers NfL, BDNF, and NSE with clinical and laboratory parameters in KT patients. * *p* < 0.05, ** *p* < 0.01, *** *p* < 0.001. Abbreviations: Apo-B—apolipoprotein–B; HDL—high density lipoprotein; Apo-A1—apolipoprotein-A1; hsCRP—high sensitivity C-reactive protein; HBA1c—hemoglobin A1c, NfL—neurofilament light chain; BDNF—brain-derived neurotrophic factor; NSE—neuron-specific enolase. Red dotted lines mark demarcation indicators for baseline and 2 years follow-up.

**Figure 3 ijms-24-06628-f003:**
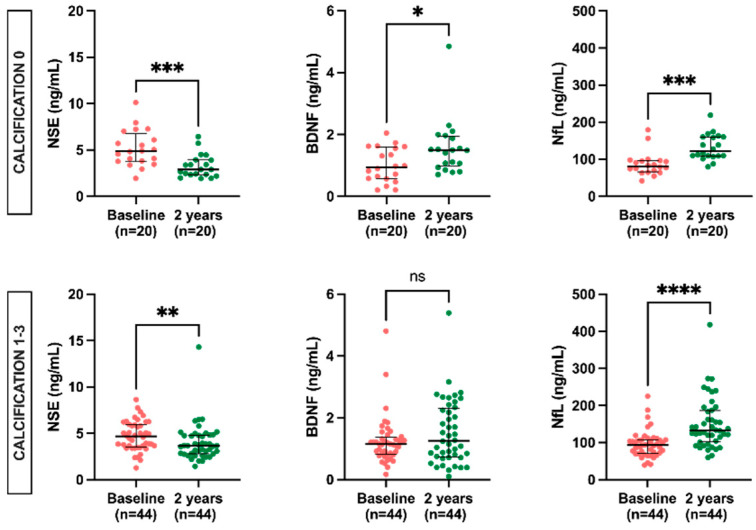
Levels of NSE, BDNF, and NfL at baseline vs. 2 years post-KT according to calcification scores. Results are presented as median and IQR. Differences between baseline and two years post-KT were analyzed using the Wilcoxon paired test. * *p* < 0.05, ** *p* < 0.01, *** *p* < 0.001, **** *p* < 0.0001.

**Figure 4 ijms-24-06628-f004:**
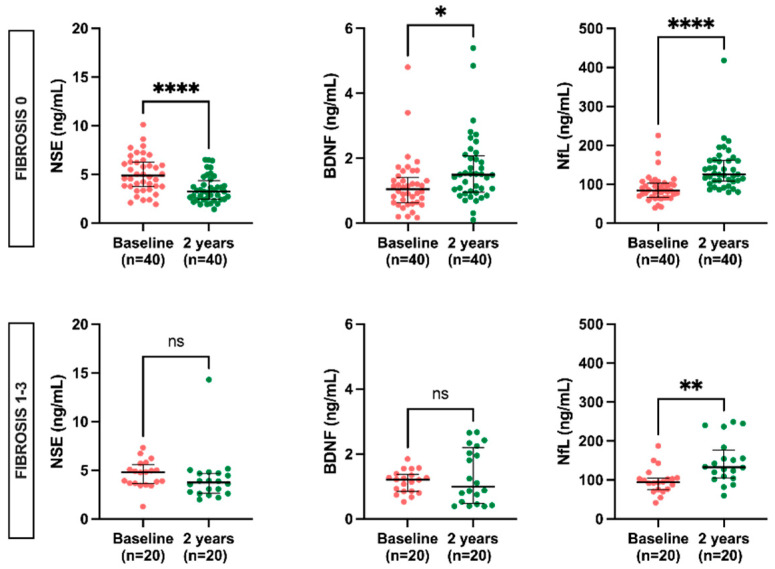
Levels of NSE, BDNF, and NfL at baseline vs. 2 years post-KT according to fibrosis score. Results are presented as the median and IQR. Differences between baseline and two years post-KT were analyzed using the Wilcoxon paired test. * *p* < 0.05, ** *p* < 0.01, **** *p* < 0.0001.

**Figure 5 ijms-24-06628-f005:**
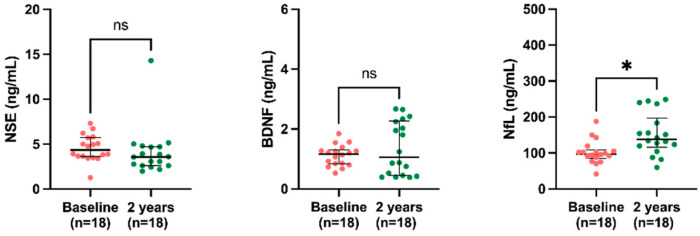
Levels of BBB biomarkers NSE, BDNF, and NfL at the baseline vs. 2 years post-KT among patients with calcification (score 1–3) and fibrosis (score 1–3). Results are presented as the median and IQR. Differences between the baseline and two years post-KT were analyzed using the Wilcoxon paired test. * *p* < 0.05.

**Figure 6 ijms-24-06628-f006:**
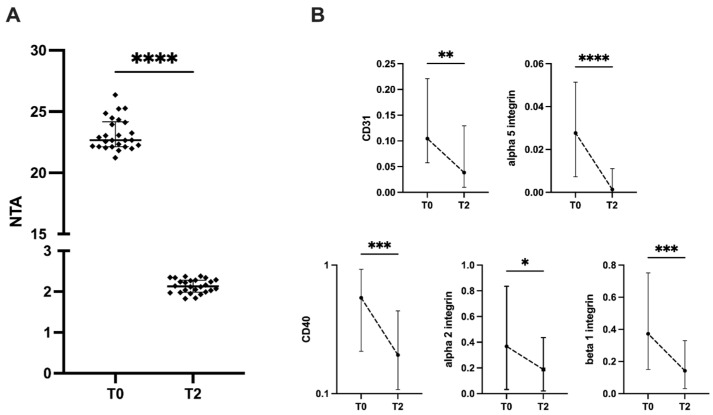
(**A**) Nanosight analysis of plasma EV concentration before (T0) and after KT (T2); (**B**) flow cytometry analysis of the surface protein expression of CD31, alpha 5 integrin, CD40, alpha 2, and beta 1 integrins. Results are presented as the median and IQR. Differences between the baseline (T0) and two years post-KT (T2) were analyzed using the Wilcoxon paired test. * *p* < 0.05, ** *p* < 0.01, *** *p* < 0.001, **** *p* < 0.0001. Abbreviations: NTA—nanoparticle tracking analysis; CD—cluster of differentiation.

**Table 1 ijms-24-06628-t001:** Clinical profile of KT patients at baseline and two years post-KT.

Parameters	Baseline (n = 74)	Two Years (n = 74)	*p* Value
Age (yrs)	46 (32–53)	48 (34–55)	-
Male Sex	52 (70)		
BMI (kg/m^2^)	24.6 (22.6–28.0), n = 31	25.3 (23.3–29.3), n = 31	0.0086 *
SBP (mmHg	138 (127–159), n = 34	132 (123–142), n = 34	0.0380 *
DBP (mmHg)	86 (77–97), n = 34	80 (77–83), n = 34	0.0794
Vintage (years)	0. 8 (0.6–3.0), n = 39	-	-
Comorbidities			
CVD	62 (84)	-	-
DM	6 (8)	-	-
Medications			
Ca Channel Blockers	40 (54)	31 (37)	0.0358 *
Beta Blockers	48 (65)	45 (54)	0.1753
ACEi/ARBs	42 (57)	62 (75)	0.0176 *
Statins	20 (27)	53 (64)	<0.0001
Biochemistry			
Creatinine, μmol/L	678 (578–854), n = 73	111 (97–136), n = 73	<0.0001 *
Albumin, g/L	35.5 (32.8–38.0), n = 70	38 (36–39), n = 70	<0.0001 *
hsCRP, mg/L	0.7 (0.3–1.7), n = 71	1.1 (0.6–2.4), n = 71	0.0414 *
Calcium, mmol/L	2. 3 (2.1–2. 4), n = 68	2.4 (2.3–2.5), n = 68	<0.0001 *
Phosphate, mmol/L	1.7 (1.3–2.0), n = 68	1.0 (0.8–1.1), n = 68	<0.0001 *
Troponin T, μg/L	21 (13–38), n = 62	8 (6–12), n = 62	<0.0001 *
Triglycerides, mmol/L	1.3 (1.0–1.8), n = 66	1.4 (1.0–1.7), n = 66	0.9272
Cholesterol, mmol/L	4. 5 (3.7–5.1), n = 66	4.4 (3.9–5.1), n = 66	0.4836
HDL-cholesterol, mmol/L	1.4 (1.0–1.7), n = 67	1.4 (1.2–1.9), n = 67	0.0069 *
Apo-A1, g/L	1.40 (1.18–1.61), n = 65	1.49 (1.31–1.77), n = 65	0.0003 *
Apo-B, g/L	0.89 (0.71–1.01), n = 64	0.77 (0.65–0.97), n = 64	0.3044
Lp(a), mg/L	51 (14–122), n = 45	12 (10–32), n = 45	<0.0001 *
HBA1c, mmol/mol	35 (32–39), n = 63	37 (35–42), n = 63	<0.0001 *
Homocysteine, μmol/L	36 (29–48), n = 66	18 (13–22), n = 66	<0.0001 *
Glucose, mmol/L	5.7 (4.9–7.40), n = 23	5.8 (5.4–6.4), n = 23	0.3160

Data are presented as the median and interquartile range (Q1–Q3). The differences between the baseline and two years post-KT were analyzed using the Wilcoxon paired test. Significance was determined at * *p* < 0.05. Abbreviations: BMI—body mass index; SBP; systolic blood pressure; DBP—diastolic blood pressure; CVD—cardiovascular disease; DM—diabetes mellitus; Ca—calcium; ACEi—angiotensin converting enzyme inhibitor; ARB—angiotensin receptor blocker; hsCRP—high sensitivity C-reactive protein; HDL—high density lipoprotein; Apo-A1—apolipoprotein-A1; Apo-B—apolipoprotein–B; Lp(a)—lipoprotein(a); HBA1c—hemoglobin A1c.

**Table 2 ijms-24-06628-t002:** Sex-divided analysis of NSE, BDNF, and NfL between male and female patients at baseline and two years post-KT.

	Baseline		*p*-Value	2 Years		*p*-Value
	Females	Males		Females	Males	
NSE, ng/mL	4.7 (3.8–5.7; n = 22)	4.5 (3.4–6.0; n = 52)	0.57	3.5 (2.5–3.9; n = 22)	3.4(2.6–4.8; n = 52)	0.44
BDNF, ng/mL	1.2 (0.8–1.6; n = 22)	1.09 (0.7–1.4; n = 52)	0.35	1.1 (0.7–1.5; n = 22)	1.5 (0.9–2.3; n = 52)	0.05 *
NfL, ng/mL	86.5 (70.2–108.2; n = 22)	94.7 (67.0–110.5; n = 52)	0.75	113.3 (99.7–144.8; n = 22)	138.2 (110.4– 173.1; n = 52)	0.07

Data are presented as median and quartile range (Q1–Q3). Statistical comparisons between sexes in each group were performed with a nonparametric Mann–Whitney U test, * *p* < 0.05. Abbreviations: NSE—neuron-specific enolase; BDNF—brain-derived neurotrophic factor; NfL—neurofilament light chain.

## Data Availability

Due to ethical reasons, the identity of individuals who took part in this study cannot be publicly shared. However, if requested, the corresponding author can provide data on reasonable grounds.

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
