# Peer review of "Blood–Brain Barrier Biomarkers before and after Kidney Transplantation"

_ijms, 2023, doi:10.3390/ijms24076628_

Round 1

Reviewer 1 Report

The manuscript is very interesting, however, I suggest correcting the following errors:

1. Laboraotry parameters measured in the paper change in time and the obtained results may depend on that. Please add more analysis or discuiss that problem.

2. Please clarify for the readers why you chose CD40, CD31, the exact types of integrins for the studies. For the unfamilliar readers the choice seems random. 

3. Adding the reasoning for the choice of each statistical test would benefit the paper :) maybe adding to materials and methods would be a good idea?

Otherwise, the paper is good and I wish the best luck in improving it. 

Author Response

Response to Reviewer 1 Comments

Point 1: Laboratory parameters measured in the paper change in time and the obtained results may depend on that. Please add more analysis or discuss that problem.

Response 1: Thank you for the feedback. We have now addressed the issue of changing laboratory parameters and the potential impact on results in the discussion section (see page 10, lines 304-329).

Point 2: Please clarify for the readers why you chose CD40, CD31, the exact types of integrins for the studies. For the unfamilliar readers the choice seems random.

Response 2: To address your concern regarding the selection of CD40, CD31, and integrins as biomarkers. As detailed in the methods section of our paper (see page 12, lines 385-386), we utilized the MACSPlex assay, which allowed for the simultaneous analysis of 39 different antigens. Our selection of CD40, CD31, and integrins alpha 2, alpha 5, and beta 1 was based on the significant differences we observed (at least < 0.05) among all the antigens tested. To clarify the rationale behind our selection of these markers for the readers, we have included an additional sentence in the method section (see page 12, lines 397-399).

Furthermore, we chose to focus on endothelial microparticles (CD31+) as they are well-established markers of vascular damage in the literature, and we have previously published on this topic in the Journal of Immunology (Cavallari et al., 2019).

We understand that the choice of biomarkers may seem random to those unfamiliar with the MACSPlex assay. However, we chose to discuss the molecules with significant results to avoid overwhelming the reader with an excessive amount of data. Additionally, we discussed the relevance of these biomarkers in the discussion section of our paper.

Thank you again for your valuable comments, and we hope that this clarifies our rationale for selecting CD40, CD31, and integrins as biomarkers in our study.

Point 3: Adding the reasoning for the choice of each statistical test would benefit the paper :) maybe adding to materials and methods would be a good idea?

Response 3: To elaborate on the reasoning for selecting the statistical tests in our study, we have now included additional information in our method section (see page 12, lines 411-417).

Reviewer 2 Report

Dear authors,

Your manuscript tackles an important and interesting topic using a strong and sound research methodology. The discussion section provides adequate explanations of the results and successfully correlates and compares them to the currently available literature. 

Before acceptance for publication, I would like to kindly ask of you to read the manuscript several times and correct the grammatical and syntax errors, as I have noticed a number of those, in order to optimize the manuscript's overall quality and credibility.

Author Response

Response to Reviewer 2 Comments

Point 1: Your manuscript tackles an important and interesting topic using a strong and sound research methodology. The discussion section provides adequate explanations of the results and successfully correlates and compares them to the currently available literature. 

Before acceptance for publication, I would like to kindly ask of you to read the manuscript several times and correct the grammatical and syntax errors, as I have noticed a number of those, in order to optimize the manuscript's overall quality and credibility

Response 1: We appreciate your valuable recommendation to review certain grammatical and syntax errors. We have carefully reviewed and revised the manuscript and have now corrected the errors. We hope our revisions have improved our manuscript’s clarity.
